# Ceftazidime-Avibactam for the Treatment of Carbapenem-Resistant Organisms: A Prospective, Observational, Single-Center Study

**DOI:** 10.3390/antibiotics14080773

**Published:** 2025-07-31

**Authors:** Frieder Pfäfflin, Anja Theloe, Miriam Songa Stegemann, Rasmus Leistner, Leif Erik Sander, Florian Kurth, Stephan Achterberg

**Affiliations:** 1Department of Infectious Diseases, Respiratory Medicine and Critical Care, Charité-Universitätsmedizin Berlin, 10117 Berlin, Germany; miriam.stegemann@charite.de (M.S.S.); leif-erik.sander@charite.de (L.E.S.); florian.kurth@charite.de (F.K.); stephan.achterberg@charite.de (S.A.); 2Antimicrobial Stewardship, Charité-Universitätsmedizin Berlin, 10117 Berlin, Germany; anja.theloe@charite.de; 3Pharmacy, Charité-Universitätsmedizin Berlin, 13353 Berlin, Germany; 4Medical Department, Division of Gastroenterology, Infectious Diseases and Rheumatology, Charité-Universitätsmedizin Berlin, 12203 Berlin, Germany; rasmus.leistner@charite.de; 5Department of Infectious Diseases, Vivantes Auguste-Viktoria-Klinikum, 12157 Berlin, Germany

**Keywords:** ceftazidime–avibactam, carbapenem-resistant organisms, antimicrobial stewardship, intensive care unit, nosocomial infection

## Abstract

**Introduction**: The World Health Organization has declared carbapenem-resistant organisms a research and development priority. Although ceftazidime–avibactam was approved around a decade ago, there is still a lack of prospective data on the treatment of resistant pathogens with this agent. **Methods**: We conducted a prospective, observational, single-center, investigator-initiated study of patients treated with ceftazidime–avibactam for infections caused by carbapenem-resistant organisms. The primary outcome was clinical cure 14 days after the initiation of ceftazidime-avibactam treatment. Secondary outcomes, which were assessed on day 30, included microbiological failure, development of resistance, all-cause mortality, and length of stay in the intensive care unit. **Results**: A total of 50 patients were included in the study. At baseline, the median Charlson Comorbidity Index and Sequential Organ Failure Assessment Score were 5.5 and 7. Approximately three-quarters of the patients were treated in an intensive care unit and had undergone mechanical ventilation within the previous 7 days prior to the commencement of ceftazidime–avibactam treatment. Half of the patients were diagnosed with nosocomial pneumonia. Most infections were caused by *Pseudomonas aeruginosa* (48%) and *Klebsiella pneumonia* (28%). Clinical cure at day 14 was achieved in 59% of patients. Four deaths (9%) and two cases of microbiological failure (4%) were observed. The median length of stay in the intensive care unit was 14 days. There was no emergence of resistance to ceftazidime–avibactam. **Discussion**: Our study contributes to the growing body of evidence supporting the effectiveness of ceftazidime–avibactam in treating infections caused by carbapenem-resistant organisms. In this cohort of critically ill patients, our results in terms of both clinical success and survival are in the upper range compared to those from mainly retrospective and some prospective studies. Although the benefits of ceftazidime–avibactam have been demonstrated in this and other studies, it must be prescribed cautiously to ensure it remains effective.

## 1. Introduction

In 2019, the World Health Organization (WHO) identified antimicrobial resistance (AMR) as one of the top ten global public health threats facing humanity [1]. The WHO also defined priority pathogens for which the research and development of new antibiotics are important. In the 2024 update, priority was given to carbapenem-resistant *Acinetobacter* (*A*.) *baumanii*, carbapenem-resistant Enterobacterales, and third-generation cephalosporin-resistant Enterobacterales [2]. Carbapenem-resistant organisms (CROs) have traditionally been treated with combinations of agents with high toxicity (aminoglycosides, colistin), suboptimal pharmacokinetics (aminoglycosides, colistin, tigecycline), and/or known microbiological resistance (carbapenems) [3]. Fortunately, several new β-lactam and β-lactam/β-lactamase inhibitor antibiotics have been developed and approved in recent years, but availability has not always been guaranteed. Meropenem–vaborbactam was licensed by the European Medicines Agency (EMA) in 2018 but has only been available since the end of 2024 in Germany. Imipenem–cilastatin–relebactam and cefiderocol have been available since 2021 but their use has been limited due to their high costs and the view that they should only be used as a last resort. In 2022, ceftolozane–tazobactam was no longer available in all markets worldwide due to a recall. This leaves ceftazidime–avibactam (C-A) as the only β-lactam/β-lactamase inhibitor that has been consistently available for the treatment of CROs in Germany in recent years.

EMA has licensed C-A for the treatment of complicated intra-abdominal infections (cIAIs), complicated urinary tract infections (cUTIs), hospital-acquired pneumonia (HAP), bacteremia associated with any of the infections listed above, and infections caused by aerobic Gram-negative organisms with limited treatment options [4]. C-A has been extensively evaluated. A systematic review and meta-analysis of randomized controlled trials (RCT) evaluating C-A versus a comparator for the treatment of any infection found no significant difference between C-A and the comparator (mostly carbapenem) for 30-day all-cause mortality, late-term mortality and clinical response [5]. Several studies have also evaluated the use of C-A in treating CRO infections. However, these are largely retrospective [6,7] and are mostly limited to *K. pneumoniae* infections [3,6,8]. In addition, a recent systematic review and meta-analysis of patients with bloodstream infections or nosocomial pneumonia found lower all-cause mortality in patients with bacteremia and improved clinical cure rates in patients with bacteremia and nosocomial pneumonia. Of note, only two of the included studies were prospective [9]. We therefore undertook a prospective, real-life study to investigate this further.

## 2. Results

### 2.1. Inclusion

We screened 92 patients who had received at least one dose of C-A. Four of these patients were minors and were therefore not considered for study inclusion. Of the remaining 88 patients, 17 were receiving C-A for off-label indications, 16 had infections that were not caused by CRO, 1 patient was participating in an interventional study, 1 patient had been treated with C-A within the previous 4 weeks, and 3 patients declined to participate. Of the remaining 50 patients, 3 patients were discharged early and could not be evaluated further, and 1 patient received C-A for <3 days. Thus, 46 patients were included in the final analysis (Figure 1).

### 2.2. Baseline Characteristics

Fifteen patients (30%) were female, the median age was 58 years and the median Charlson Comorbidity Index (CCI) was 5.5. The majority of patients (77%) were treated in an intensive care unit (ICU). More than 60% of patients had cardiac and pulmonary comorbidities. A substantial proportion of patients had undergone mechanical ventilation ≤ 7 days and/or surgery ≤ 30 days prior to the start of C-A (76 and 52%, respectively). Three patients were in septic shock. Renal replacement therapy on day 1 was administered in 16 patients (32%), and extracorporeal membrane oxygenation was administered in 2 (4%) patients. The median sequential organ failure assessment (SOFA) score at baseline was 7 (Table 1).

### 2.3. Infection

The most common type of infection was nosocomial pneumonia, including ventilator-associated pneumonia (50%), followed by abdominal infections (28%, all of them complicated abdominal infections) and primary bloodstream infections (16%). CROs were primarily cultured from respiratory secretions including bronchoalveolar lavage (48%), from intraoperative specimens (30%), and from blood culture (22%). Most infections were caused by *Pseudomonas* (*P.*) *aeruginosa* and *K. pneumoniae* with a share of 48% and 28% of patients, respectively. One infection was caused by an intrinsically carbapenem-resistant organism (*Stenotrophomonas* (*S.*) *maltophilia*). Testing for carbapenemases was performed on all but one patient with Enterobacterales infections (in whom the pathogen had been isolated in a different hospital). OXA-48 and KPC were detected in 14 and 5 isolates, respectively. No carbapenemases were found in five CR Enterobacterales isolates (one isolate each showed combined porin loss and ESBL, and combined porin loss and AmpC; no resistance genes were detected in three isolates). The median (IQR) minimum inhibitory concentration (MIC) of C-A for *P. aeruginosa* was 4 (3.75–4.5); the median (IQR) MIC of C-A for all other CROs was 1 (1–2).

C-A was administered for a median duration of 13 days (assessed at day 30). Combination treatment (i.e., additional treatment with drugs to which the isolate was susceptible) was given to 23 (45%) patients. Partner drugs were administered intravenously only, nebulized only, or both intravenously and nebulized in 10, 8, and 5 patients, respectively. Polymicrobial infections were frequent. Additional pathogens were detected in 35 (69%) patients, most commonly *Candida* spp., *E. faecium*, and coagulase-negative staphylococci, occurring in 14, 11, and 10 patients, respectively. Multidrug therapy (i.e., additional treatment with drugs to which the isolate was not susceptible) was administered for 37 (80%) patients. Echinocandins were administered most frequently (17 patients), followed by linezolid, vancomycin, and metronidazole in 12, 10, and 8 patients, respectively (Table 2).

### 2.4. Primary Outcome

Primary and secondary outcomes could be assessed in 46 patients. Clinical cure was achieved in 27 patients (59%). The reasons for failure were death in three (7%) patients, continued need for vasopressors in two (4%) patients, renewed detection of CROs from primarily sterile sites after ≥7 days of treatment in two (4%) patients, and no improvement in the oxygenation index in two (4%) patients. Additionally, the SOFA score was not improved in 10 (22%) patients (Table 3).

### 2.5. Secondary Outcomes

The index isolate was detected in two patients from primarily sterile sites ≥ 7 days after the start of treatment. Both were patients with complicated abdominal infections in whom the infection focus could not be optimally sanitized. In addition, in five patients, index isolates were cultured from respiratory samples and were considered to be colonizers. None of the isolates, all of which had been susceptible in the first round, had become resistant to C-A in the second antimicrobial susceptibility test. The overall mortality at day 30 was 8.7%. The median length of stay (LOS) in the ICU (calculated on day 30) was 14 days (Table 3).

## 3. Discussion

In this prospective, single-center, observational study of patients infected with CRO, we demonstrate that 59% of patients achieved the composite primary outcome of clinical cure at day 14. Our cohort mostly comprised middle-aged patients (median age 58 years) with a high CCI (median 5.5). Around three-quarters of the patients had both been treated in the ICU and mechanically ventilated in the last seven days before study inclusion. The most common infections were nosocomial pneumonia and intra-abdominal infections (50 and 28%, respectively). The predominant carbapenem-resistant pathogens were *P. aeruginosa* (48%) and *K. pneumoniae* (28%). All-cause mortality at day 30 was low (9%).

While clinical cure by day 14 in just over half of the patients may seem like a modest result, we applied stringent criteria for this composite outcome based on the AIDA study by Paul et al., in which clinical failure at day 14 was observed in 79% of patients, albeit in a somewhat different patient population [10]. The median SOFA score on day 1 was 6 in their study, compared to 7 in ours. Importantly, however, the majority of infections (77%) were caused by *A. baumanii* with a high MIC for meropenem, and were treated with colistin alone or in combination with meropenem. Active β-lactam antibiotics were unavailable, which may explain the poorer clinical outcome. A multicenter retrospective study of 105 patients who were treated with C-A for infections caused by CROs observed clinical success at day 30 in 61% of patients [11], similar to our findings at day 14. The criteria for clinical success in that study were less rigorous than ours (e.g., unlike in our study, the researchers did not require stable vital signs, improvements in the SOFA score, or improvements in the oxygenation index).

The all-cause in-hospital mortality rate on day 30 in our study (4/46, 8.7%) is comparable to the results of a prospective study by van Duin et al., in which the treatment of infections due to CROs (almost exclusively *K. pneumoniae*) with colistin or C-A was investigated, and an all-cause in-hospital mortality rate of 8% (3/38) was similarly observed in patients treated with C-A [8]. To date, the largest prospective study—a national registry study from Greece—included patients infected mainly by KPC-producers and found a 28-day mortality rate of 20% [12]. Similar results to the latter study have been observed in retrospective investigations: a recent retrospective multicenter study of patients with carbapenem-resistant *K. pneumoniae* bloodstream infections found a 30-day mortality rate of 34%, which was reduced to 21% in patients who were appropriately treated with C-A [7]. The main differences compared to our study were that the aforementioned study had an exclusive focus on bacteremic *K. pneumoniae* infections and a multicenter retrospective design. Another observational retrospective study including 171 patients treated with C-A identified 30-day mortality in 22% of patients. All infections were due to Enterobacterales harboring OXA-48 [13]. We did not observe the development of resistance to C-A in our patients, although estimates of the emergence of resistance to C-A after exposure range from 10 to 20% [14,15]. However, resistance rates may be in the low single digits, as found in larger observational studies [11,12].

Approximately half of our patients had infections caused by carbapenem-resistant *P. aeruginosa*. Interestingly, a recent multicenter, retrospective, observational study compared ceftolozane–tazobactam with C-A in the treatment of multidrug-resistant *P. aeruginosa* (i.e., the isolate was non-susceptible to at least one agent in three or more antibiotic classes). Similar proportions of patients to those in our study were recruited from the ICU and were mechanically ventilated. The primary outcome of clinical success at day 30 was observed in 61% and 52% of patients who were treated with ceftozolane–tazobactam and C-A, respectively [16]. The emergence of resistance was common with both agents during the 90-day observation period (22%). Although we did not observe any resistance to C-A in our study, this was probably due to our shorter observation period of 30 days. Ceftolozane–tazobactam has been licensed in Germany since 2015. This drug is of great importance in the treatment of infections caused by multidrug-resistant *P. aeruginosa*. However, C-A has a broader spectrum of activity against Gram-negative bacteria harboring carbapenemases.

Our study has several limitations. Firstly, our study was small and single-center, so the findings may not be generalizable. Secondly, the study lacked a control group. We had originally planned to include matched historical controls but this was not feasible (see details in the methods section). Thirdly, the landscape of carbapenemases changes over time. OXA-48 has been the predominant carbapenemase in Enterobacterales in Germany and still is, according to a recent report from the German national reference center for Gram-negative bacteria [17]. However, in many areas, CROs susceptible to C-A have been replaced by organisms carrying metallo-β-lactamases. In Germany, this has particularly been reported for *K. pneumoniae* in patients with exposure in Ukraine [18]. C-A is the preferred treatment option for infections caused by OXA-48-producing CROs, whereas for pathogens carrying metallo-β-lactamases, preferred treatment options include C-A in combination with aztreonam or cefiderocol as monotherapy [14]. Alternatively, the recently approved combination of aztreonam and avibactam can be used to treat infections caused by aerobic Gram-negative pathogens for which there are limited treatment options [19].

The global burden of bacterial antimicrobial resistance is increasing. Among the Gram-negative bacteria, resistance to carbapenems has increased more than resistance to any other class of antibiotic, rising from 127,000 attributable deaths in 1990 to 216,000 in 2021. Forecasts indicate that antimicrobial resistance will lead to an even greater number of deaths in the future [20]. Carbapenem-resistant Enterobacterales have been declared a top priority for research and development by the WHO [2]. Antimicrobial stewardship (AMS) programs are critical in improving antibiotic selection and dosing and optimizing treatment duration to keep CROs in check [21]. It has been shown that AMS programs can lead to a sustained reduction in antibiotic use and a decreased incidence of nosocomial infections and CRO colonization [22]. Reserve antibiotics such as C-A are important therapeutic options for CRO-related infections. While the benefits of these antibiotics have been demonstrated in this and other studies, they must be prescribed cautiously to maintain their effectiveness.

## 4. Methods

Charité-Universitätsmedizin Berlin, Germany is a large tertiary care hospital with more than 3000 in-patient beds and 25 ICUs across three campuses. As of May 2025, the AMS team consisted of six physicians (one microbiologist and five infectious disease physicians) and two clinical pharmacists. This study was an investigator-initiated, single-center, prospective, observational study. The study was approved by the Ethics Committee of Charité-Universitätsmedizin Berlin (application number EA2/251/18). Patients were included if they were ≥18 years of age and had received ≥1 dose of C-A for a documented infection with a CRO. Patients were excluded if they were included in interventional trials, if they had been treated with C-A for ≥3 days within ≤4 weeks, or if C-A had been administered off-label (off-label use was not covered by the ethics committee vote). Only patients who had received C-A for a duration of ≥3 days were included in the final analysis.

We hypothesized that C-A is more effective than other antibiotics in treating CRO infections. We had originally planned to use a matched historical control group of patients with infections due to CROs who had been treated prior to the availability of C-A. However, patient recruitment was slow during the Coronavirus Disease 2019 (COVID-19) pandemic. In the meantime, a new version of our patient data management system was implemented and the data of historical controls could no longer be accessed.

### 4.1. Outcomes

The primary outcome was clinical cure 14 days after initiation of C-A. Clinical cure was defined as a composite of all of the following criteria: patient alive, systolic blood pressure ≥ 90 mmHg without vasopressors, improvement in the SOFA score (for a baseline SOFA score < 3, the score had to remain stable or decrease), and, for pneumonia improvement or the stability of the oxygenation index, microbiological cure (no detection of the index isolate from primarily sterile sites ≥ 7 days after start of treatment).

Secondary outcomes included microbiological failure (detection of the index isolate from primarily sterile sites ≥ 7 days after initiation of treatment), development of resistance to C-A, all-cause mortality at day 30, and LOS in the ICU.

Patients were followed if they received ≥3 days of treatment with C-A until 30 days after the initiation of therapy or until discharge, transfer to another hospital, death, or loss to follow-up, whichever occurred first.

### 4.2. Statistical Analysis

Univariate, descriptive analysis was used for data analysis. Patient characteristics are presented as absolute frequencies. Metric and ordinal numbers are presented with medians and interquartile ranges (IQR).

Retrospective studies have shown that clinical success with C-A in patients with CRO infections was at least 30% higher than that with other antibiotics. Based on these data, we initially calculated that we would need to include 38 patients to achieve 80% power and a two-sided significance of 5%. We anticipated losing 20% of the patients by the end of the follow-up period on day 30 after the start of treatment. Therefore, 46 patients and—with a matching ratio of 1:2—92 controls had to be included. However, the latter could not eventually be included.

## Figures and Tables

**Figure 1 antibiotics-14-00773-f001:**
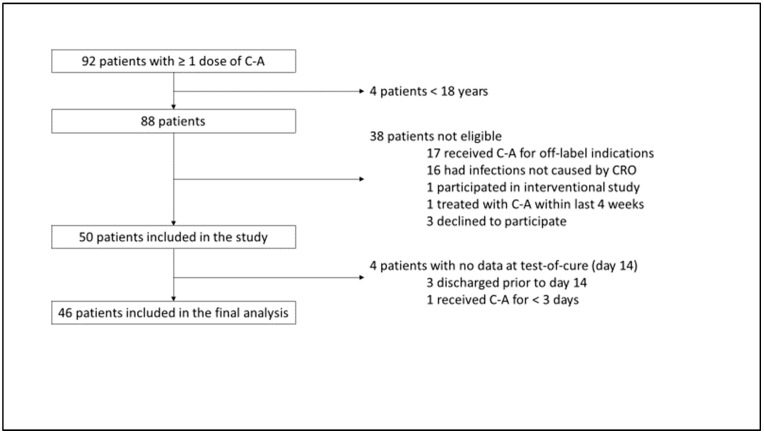
Study tree. C-A = ceftazidime–avibactam; CRO = carbapenem-resistant organism.

**Table 1 antibiotics-14-00773-t001:** (a) Baseline characteristics; (b) clinical and laboratory parameters at baseline and test-of-cure.

Demographics and Background	No of Patients (%) If Not Specified Otherwise	
Patients analyzed	50	
Female	15 (30)	
Age [y], median (IQR)	58 (51–71)	
Charlson Comorbidity Index, median (IQR)	5.5 (3–8)	
Type of ward at day 1		
General medical ward	9 (18)	
General surgical ward	2 (4)	
Medical ICU	29 (57)	
Surgical ICU	10 (20)	
Main comorbidities		
Cardiac	32 (64)	
Chronic pulmonary disease	31 (62)	
Malignancy	14 (28)	
Chronic kidney disease	16 (32)	
Diabetes	13 (26)	
Chronic liver disease	7 (14)	
Solid organ transplantation	5 (10)	
HIV	2 (4)	
Immunosuppression		
Prednisolone equivalent ≥ 20 mg/d for ≥14 d within last mth	8 (16)	
Other immunosuppressants for ≥1 mth within last 6 mths	9 (18)	
Chemotherapy within last 6 mths	4 (8)	
Neutropenia within 7 d prior to study inclusion	1 (2)	
Invasive ventilation within 7 d prior to study inclusion	38 (76)	
Surgery within last mth prior to study inclusion	26 (52)	
	Status at day 1	Status at day 14
Septic shock	3 (6)	-
Invasive ventilation	32 (64)	24 (52)
Non-invasive ventilation	9 (18)	10 (22)
ECMO	2 (4)	1 (3)
Intermittent hemodialysis	4 (8)	4 (8)
Continuous veno-venous hemodiafiltration	12 (24)	7 (15)
SOFA score, median (IQR)	7 (4–11)	5 (2–8)
Systolic blood pressure [mmHg], median (IQR)	115 (110–122)	115 (110–120)
Oxygenation index, median (IQR)	286 (205–393)	344 (288–424)
Creatinine [mg/dL], median (IQR)	1.0 (0.7–1.7)	1.0 (0.6–1.7)
Total bilirubine [mg/dL], median (IQR)	0.8 (0.4–2.0)	0.5 (0.3–1.2)
ALT [U/L], median (IQR)	37 (22–67)	31 (14–58)
Leukocytes [/nL], median (IQR)	14 (9–18)	10 (8–14)
CrP [mg/L], median (IQR)	155 (107–188)	90 (35–148)
PCT [µg/L], median (IQR)	1.0 (0.4–2.4)	0.8 (0.3–1.7)

(a) Baseline demographic and background data, type of ward, main comorbidities, immunosuppression, ventilation and recent surgery; (b) baseline and test-of-cure (day 14) clinical and laboratory parameter data. Data are presented as n (%) or median (IQR). IQR = interquartile ranges, ICU = intensive care unit, HIV = human immunodeficiency virus, ECMO = extracorporeal membrane oxygenation, SOFA = sequential organ failure assessment, ALT = alanine transaminase, CrP = C-reactive protein, and PCT = procalcitonin.

**Table 2 antibiotics-14-00773-t002:** Infection characteristics and antimicrobial treatment.

	No of Patients (%) If Not Specified Otherwise	No of Isolates with Carbapenemases
Type of infection *		
Hospital acquired pneumonia	25 (50)	
Abdominal infection	14 (28)	
Blood stream infection	8 (16)	
Skin and soft tissue infection	3 (6)	
Complicated urinary tract infection	1 (2)	
Bone infection	1 (2)	
Type of specimen #		
Respiratory secretion	24 (48)	
Intraoperative sample	15 (30)	
Blood culture	11 (22)	
Urin	1 (2)	
Other	1 (2)	
Carbapenem-resistant pathogens		Carbapenemases
*P. aeruginosa*	24 (48)	n.a.
*K. pneumoniae*	14 (28)	OXA-48 (10), KPC (3)
*K. aerogenes*	4 (8)	KPC (1)
*E. coli*	3 (6)	OXA-48 (3)
*E. cloacae complex*	3 (6)	OXA-48 (1)
*S. maltophilia*	1 (2)	n.a.
*C. freundii*	1 (2)	KPC (1)
*S. marcescens*	1 (2)	None
Other pathogens in polymicrobial infection	*Candida* spp. 14 *E. faecium* 11 Coagulase-negative staphylococci 10 *P. aeruginosa* 4 *S. maltophilia* 4 *K. pneumoniae* 3 *E. coli* 3 *Proteus* spp. 3 *K. aerogenes* 2 *E. cloacae* 2 *E. faecalis* 1 *R. planticola* 1 *C. koseri* 1 *S. marcescens* 1 *E. durans* 1 *S. aureus* 1 *P. distasonis* 1 *A. flavus* 1 *Fusarium* sp. 1	
Treatment		
Treatment duration with C-A [d], median (IQR)	13 (10–18)	
Combination antibiotic therapy	23 (45)	
Intravenous partner drugs	Colistin 4 Ciprofloxacin 4 Gentamicin 4 Tigecycline 1 Aztreonam 1 Fosfomycin 1 Cotrimoxazole 1	
Nebulized partner drugs	Colistin 10 Tobramycin 3 Gentamycin 2	
Multidrug therapy	11 (22)	
Antibacterial drugs in multidrug regimens	Linezolid 12 Vancomycin 10 Metronidazole 8 Meropenem 7 Cotrimoxazole 6 Ampicillin-sulbactam 4 Ampicillin 4 Daptomycin 3 Imipenem 2 Ciprofloxacin 2 Tigecycline 2 Rifampicin 1 Teicoplanin 1 Fosfomycin 1	
Antifungal drugs in multidrug regimens	Caspofungin 15 Liposomal amphotericin B 3 Anidulafungin 2 Voriconazole 2 Fluconazole 1	

Type of infection, type of specimen, species of carbapenem-resistant pathogen, presence of other pathogens, and antimicrobial treatment are displayed. Carbapenem-resistant pathogens are index isolates that enabled study inclusion and were treated with C-A. Other pathogens were cultured in polymicrobial infections. Partner drugs were administered in combination with C-A to treat the index isolates. Drugs in multidrug regimens were used to treat the other pathogens. * Nosocomial pneumonia was diagnosed simultaneously with bloodstream infection and skin and soft tissue infections in one patient each. # The index pathogen was isolated from respiratory specimens at the same time as from intraoperative specimens or blood cultures in one patient each. C-A = ceftazidime–avibactam, *P. aeruginosa* = *Pseudomonas aeruginosa*, *K. pneumoniae* = *Klebsiella pneumoniae*, *K. aerogenes* = *Klebsiella aerogenes*, *E. coli* = *Escherichia coli*, *E. cloacae* = *Enterobacter cloacae*, *S. maltophilia* = *Stenotrophomonas maltophilia*, *C. freundii* = *Citrobacter freundii*, *S. marcescens* = *Serratia marcescens*, *E. faecium* = *Enterococcus faecium*, *E. faecalis* = *Enterococcus faecalis*, *R. planticola* = *Raoultella planticola*, *C. koseri* = *Citrobacter koseri*, *E. durans* = *Enterococcus durans*, *S. aureus* = *Staphylococcus aureus*, *P. distasonis* = *Parabacteroides distasonis*, *A. flavus* = *Aspergillus flavus*, and n.a. = not applicable.

**Table 3 antibiotics-14-00773-t003:** Outcomes.

	No of Patients (%) If Not Specified Otherwise
Primary outcome (day 14)	
Clinical cure	27 (59)
Reasons for clinical failure	Death 3 (7) Vasopressors 2 (4) CROs after treatment ≥ 7 d 2 (4) Oxygenation index not improved 2 (4) SOFA not improved 10 (22)
Secondary outcomes (day 30)	
All-cause mortality at day 30	4 (9)
Length of stay in ICU [d], median (IQR)	14 (2–25)
Detection of CROs after treatment ≥ 7 d	2 (4)
Resistance to C-A	0

Primary outcome at test-of-cure (day 14) and secondary outcomes (day 30). Clinical cure was defined as a composite of all of the following criteria: patient alive, systolic blood pressure ≥ 90 mmHg without vasopressors, improvement in the SOFA score (for a baseline SOFA score < 3, the score had to remain stable or decrease), and, for pneumonia improvement or stability of the oxygenation index, microbiological cure (no detection of the index isolate from primarily sterile sites ≥ 7 days after start of treatment). C-A = ceftazidime–avibactam, ICU = intensive care unit, LOS = length of stay, CRO = carbapenem-resistant organism, and SOFA = sequential organ failure assessment.

## Data Availability

The raw data supporting the conclusions of this article will be made available by the authors on request.

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
