# Peer review of "Ceftazidime-Avibactam for the Treatment of Carbapenem-Resistant Organisms: A Prospective, Observational, Single-Center Study"

_antibiotics, 2025, doi:10.3390/antibiotics14080773_

Round 1

Reviewer 1 Report

Comments and Suggestions for Authors

Frieder Pfäfflin et al have conducted a prospective study of patients treated with 24 ceftazidime-avibactam for infections caused by carbapenem-resistant organisms. The primary outcome was clinical cure 14 days after the initiation of ceftazidime-avibactam treatment. They have found that clinical cure at day 14 was achieved in 59% of cases.

I have the following comments and suggestions:

ABSTRACT

Please put the Conclusions after Results not Discussion in the abstract

INTRODUCTION

The second paragraph of introduction is very extensive and part should be transferred in discussion section

As the study is prospective a study hypothesis should be stated before the authors describe the aim of this study

PATIENTS AND METHODS

Please write patient instead of cases

GENERAL COMMENTS

I cannot understand well what the prospective nature of this study offers to the improvement comparing with a similar retrospective study.

In general, the study actually describes the local conditions of an ICU that is interesting but it is so long to be followed well by the reader.

We believe that a short report containing not more than 2000 words and 1-2 tables would be more attractive and convincing to the reader. The issue is interesting and the clinical experience of these valuable drugs should be disseminated so I counsel the author to review and shortened considerably their manuscript before acceptance. THE information that is given but this report is not so large to cover a large paper with lots of tables and details of questionable interest

Author Response

Dear Reviewer 1,

Thank you very much for your review and your comments and suggestions. They were very helpful. We agreed in most issues and have tried to adapt the manuscript accordingly. You may find our modifications detailed below and also in the manuscript. We hope that the amended manuscript will meet with your approval.

Yours sincerely

Frieder Pfäfflin

Frieder Pfäfflin et al have conducted a prospective study of patients treated with 24 ceftazidime-avibactam for infections caused by carbapenem-resistant organisms. The primary outcome was clinical cure 14 days after the initiation of ceftazidime-avibactam treatment. They have found that clinical cure at day 14 was achieved in 59% of cases.

I have the following comments and suggestions:

ABSTRACT

Please put the Conclusions after Results not Discussion in the abstract

Answer: We are sorry to not fully understand this comment. The abstract is subdivided in introduction, methods, results, and discussion. We did not provide a conclusion. The last phrase of the abstract is “Although the benefits of ceftazidime-avibactam have been demonstrated in this and other studies; it must be prescribed cautiously to ensure it remains effective.” We do not consider this sentence to be a conclusion, but rather an appeal and outlook. Therefore, we believe it should rather stay in its current place.

INTRODUCTION

The second paragraph of introduction is very extensive and part should be transferred in discussion section

Answer: Thank you for this helpful comment. We agree that the content of this paragraph is extensive. We have therefore tried to summarize the main points and have considerably shortened the introduction. The passage now reads (page 2, lines 75-96): “Several studies have also evaluated the use of C-A in treating CRO infections. However, these are largely retrospective [6, 8] and are mostly limited to K. pneumoniae infections [3, 6, 7]. In addition, a recent systematic review and meta-analysis of patients with bloodstream infections or nosocomial pneumonia found lower all-cause mortality in patients with bacteremia and improved clinical cure rates in patients with bacteremia and nosocomial pneumonia. Of note, only two of the included studies were prospective [9]. We therefore undertook a prospective, real-life study to investigate this further.” 

As the study is prospective a study hypothesis should be stated before the authors describe the aim of this study

Answer: Thank you for this important comment. We have adapted the methods section. The passage now reads (page 3, lines 110-115): “We hypothesized that C-A is more effective than other antibiotics in treating CRO infections. We had originally planned to use a matched historical control group of patients with infections due to CRO who had been treated prior to the availability of C-A. However, patient recruitment was slow during the Coronavirus Disease 2019 (COVID-19) pandemic. In the meantime, a new version of our patient data management system was implemented and the data of historical controls could no longer be accessed.”

PATIENTS AND METHODS

Please write patient instead of cases

Answer: Thank you. We have adapted the manuscript accordingly.

GENERAL COMMENTS

I cannot understand well what the prospective nature of this study offers to the improvement comparing with a similar retrospective study.

In general, the study actually describes the local conditions of an ICU that is interesting but it is so long to be followed well by the reader.

We believe that a short report containing not more than 2000 words and 1-2 tables would be more attractive and convincing to the reader. The issue is interesting and the clinical experience of these valuable drugs should be disseminated so I counsel the author to review and shortened considerably their manuscript before acceptance. THE information that is given but this report is not so large to cover a large paper with lots of tables and details of questionable interest

Answer: Thank you for your comments and your concerns. We agree that our main findings are not surprising. The effectiveness of ceftazidime-avibactam hast been shown in other studies as well. However, we believe to provide additional relevant insights. This is, to our knowledge, the second largest prospective study evaluating ceftazidime-avibactam in treating CRO infections. Prospective studies in general allow to define and apply outcome criteria a priori and collect all relevant variables. The patients included in our study were comparatively sick with a high Charlson Comorbidity Index and approximately three-quarters of the patients were treated in intensive care units and had undergone mechanical ventilation in the previous seven days prior to the commencement of ceftazidime-avibactam treatment. Still, our results in terms of clinical success are in the upper range compared to other studies. We provide detailed information concerning the complicated infections in the intensive care units. Of note, this includes extensive data on other pathogens in polymicrobial infections and antimicrobials in multidrug regimens. However, we agree that there is room to shorten the manuscript. We have considerably shortened the introduction as you suggested and we have also omitted two passages from the discussion (page 10, lines 292-296; page 10-11, lines 307-313). However, we have not shortened the manuscript to about 2,000 words. We suggest this decision may be left to the editor.

Reviewer 2 Report

Comments and Suggestions for Authors

This study shows the real-world experience of using ceftazidime-avibactam for treatment of infections caused by carbapenem resistant Gram-negative bacteria. The study is interesting and expands the available information, some area of manuscript need revision prior to publication. Please see the following comments:

  1. Controls are missing from the study design.
  2. Adding the MICs of isolates for drugs (carbapenems) used for the treatment would improve the manuscript, since the study isolates were resistant to carbapenems.
  3. Line 165: Dis authors looked for other carbapenems except OXA and KPC in the isolates that didn’t harbor either of them.
  4. Did you have details on OXA and KPC variants as not all variants have the same carbapenem hydrolytic profile.
  5. Table 2: Put the values in bracket as n=10 or 3
  6. It would be interesting to know the susceptibility profile of those two CRO isolates recovered after 7 days of C-A treatment. What was MIC for C-A for those two isolates.
  7. The discussion can be shorted as some parts are overlapping with introduction (288-293).

Author Response

Dear Reviewer 2,

Thank you very much for your review and your comments and suggestions. They were very helpful. We agreed in all issues you raised and have adapted the manuscript accordingly. You may find our modifications detailed below and also in the manuscript. We hope that the amended manuscript will meet with your approval.

Yours sincerely

Frieder Pfäfflin

This study shows the real-world experience of using ceftazidime-avibactam for treatment of infections caused by carbapenem resistant Gram-negative bacteria. The study is interesting and expands the available information, some area of manuscript need revision prior to publication. Please see the following comments:

  1. Controls are missing from the study design.

Answer: Thank you for this important comment. We had intensely discussed whether our intention to include matched historical controls should be mentioned in the study design. We had omitted this section as we thought this might distract the readers from the findings. However, we agree that the study design should be described as it was planned from the beginning. We have therefore included two paragraphs in the methods section. The passages now read:

Page 3, lines 110-115: “We hypothesized that C-A is more effective than other antibiotics in treating CRO infections. We had originally planned to use a matched historical control group of patients with infections due to carbapenem-resistant organisms who had been treated prior to the availability of ceftazidime-avibactam. However, patient recruitment was slow during the Coronavirus Disease 2019 pandemic. In the meantime, a new version of our patient data management system was implemented and the data of historical controls could no longer be accessed.”

Page 3-4, lines 134-140: ”Retrospective studies have shown that clinical success with C-A in patients with CRO infections was at least 30% higher than with other antibiotics. Based on these data, we initially calculated that we would need to include 38 patients to achieve 80% power and a two-sided significance of 5%. We anticipated losing 20% of the patients by the end of the follow-up period on day 30 after start of treatment. Therefore, 46 patients and – with a matching ratio of 1:2 – 92 controls had to be included. However, the latter could not eventually be included.”

To avoid duplication, we have adapted the paragraph in the discussion. The passage now reads (page 10, lines 290-292: “We had originally planned to include matched historical controls but this was not feasible (see details in the methods section).”

  1. Adding the MICs of isolates for drugs (carbapenems) used for the treatment would improve the manuscript, since the study isolates were resistant to carbapenems.

Answer: Thank you very much for this important comment. All isolates (except for S. maltophilia, in which no carbapenem testing is performed) were tested as resistant to meropenem (i.e. meropenem MIC ≥ 16 mg/L). The actual MIC is not determined as carbapenems are not considered to be effective treatment options. The MIC of C-A were tested in all Carbapenem-resistant isolates. The median MIC (IQR) for P. aeruginosa was 4 (3.75 – 4.5), the median MIC (IQR) for all other CRO was 1 (1 – 2). We have included this information in the results section. The passage now reads (page 6, lines 183-185): “The median (IQR) minimum inhibitory concentration (MIC) of C-A for P. aeruginosa was 4 (3.75 – 4.5), the median (IQR) MIC of C-A for all other CRO was 1 (1 – 2).”

  1. Line 165: Dis authors looked for other carbapenems except OXA and KPC in the isolates that didn’t harbor either of them.

Answer: Thank you for this question. The approach in our center is to first look for carbapenemases by rapid diagnostic test. If results are negative the isolate is sent to the national reference laboratory for multidrug-resistant gram-negative bacteria for further testing (including testing for other carbapenemases). We had five Enterobacterales isolates without detection of carbapenemases. In three isolates (one isolate each of S. marcescens, K. aerogenes, and E. cloacae) no resistance genes could be identified. In one isolate of K. pneumoniae a combination of porin loss and ESBL was detected, and in one isolate of K. aerogenes a combination of porin loss and AmpC was detected. We have included this information in the manuscript (page 6, lines 181 – 183). The passage now reads: “No carbapenemases were found in 5 CR Enterobacterales isolates (one isolate each showed combined porin loss and ESBL, and combined porin loss and AmpC; no resistance genes were detected in 3 isolates).

  1. Did you have details on OXA and KPC variants as not all variants have the same carbapenem hydrolytic profile.

Answer: Thank you for this important comment. Indeed, carbapenemases differ substantially. Unfortunately, this information is not provided to us. We receive reports from the microbiology department including their own test results and also the results from the national reference laboratory. However, they only report “KPC” and do not give information whether this is KPC-1, KPC-2, or KPC-3.

  1. Table 2: Put the values in bracket as n=10 or 3

Answer: Thank you for this comment. We have adapted the table accordingly (page 7, table 2).

  1. It would be interesting to know the susceptibility profile of those two CRO isolates recovered after 7 days of C-A treatment. What was MIC for C-A for those two isolates.

Answer: Thank you for this question. The CRO isolates, which were recovered after 7 days of treatment with C-A, were both from patients with complicated abdominal infections in whom the infection focus could not be optimally sanitized. In one isolate from blood culture, the initial MIC for C-A was 2 and dropped to 1 mg/L on repeated testing from blood culture 10 days after start of treatment. The other persistent isolate was from intraabdominal specimens. The MIC increased from 4 (initial testing) to 8 mg/L after 8 days of C-A. A summary of these findings, including also the persistent colonizers, is in the manuscript (page 9, lines 232 – 234: “None of the isolates, all of which had been susceptible in the first round, had become resistant to C-A in the second antimicrobial susceptibility testing.”). We did not include these MIC values in the manuscript as there was no substantial trend and we considered this passage would become too lengthy.

  1. The discussion can be shorted as some parts are overlapping with introduction (288-293).

Answer: Thank you for your thorough review. We agree that there is some redundancy. We have therefore omitted lines 289 – 295. This includes omission of reference 20, which had only been necessary to back the predominance of KPC in the United States.